# Perception of Attitudes of the General Population toward Homosexuality in Taiwan: Roles of Demographic Factors, Mental Health, and Social Debates on Legalizing Same-Sex Marriage

**DOI:** 10.3390/ijerph18052618

**Published:** 2021-03-05

**Authors:** Huang-Chi Lin, Yi-Lung Chen, Nai-Ying Ko, Yu-Ping Chang, Wei-Hsin Lu, Cheng-Fang Yen

**Affiliations:** 1Department of Psychiatry, School of Medicine, College of Medicine, Kaohsiung Medical University, Kaohsiung 80708, Taiwan; cochigi@kmu.edu.tw; 2Department of Psychiatry, Kaohsiung Medical University Hospital, Kaohsiung 80708, Taiwan; 3Department of Healthcare Administration, Asia University, Taichung 41354, Taiwan; elong@asia.edu.tw; 4Department of Psychology, Asia University, Taichung 41354, Taiwan; 5Department of Nursing, College of Medicine, National Cheng Kung University, Tainan 70101, Taiwan; nyko@mail.ncku.edu.tw; 6School of Nursing, The State University of New York, University at Buffalo, Buffalo, NY 14260, USA; yc73@buffalo.edu; 7Department of Psychiatry, Ditmanson Medical Foundation Chia-Yi Christian Hospital, Chia-Yi City 60002, Taiwan

**Keywords:** attitude, homosexuality, same-sex marriage, mental health

## Abstract

The aims of this online survey study were to examine the associations of demographic characteristics (gender, age, and sexual orientation), mental health status, and point in time of the survey (i.e., the beginning of the social debates on legalizing same-sex marriage vs. the end of the social debates) with people’s perception of the attitudes of the general population in Taiwan toward homosexuality. A two-wave internet survey was conducted using Facebook to gather information regarding people’s perception of the population’s attitudes toward homosexuality among 4562 participants. The five-item Brief Symptom Rating Scale was used for assessing mental health status. The results indicated that participants perceived the population as having a lower acceptance at the end of the social debates on legalizing same-sex marriage than at the beginning of the social debates; gender moderated the decline in perceived acceptance. The results also indicated that poor mental health and heterosexual orientation were significantly associated with a lower perception of the population’s homosexuality acceptance in both waves of the survey. The factors related to perceived homosexuality acceptance need to be considered in developing programs to increase the population’s homosexuality acceptance.

## 1. Introduction

### 1.1. Perceived Attitudes toward Homosexuality in General Population

Lesbian, gay, and bisexual (LGB) individuals may experience social stigma in their day-to-day life [1]. Prejudice and explicitly stigmatizing behaviors of other people may increase LGB individuals’ fear regarding discrimination [2,3]. According to the minority stress model, homophobic harassment and discrimination rooted in a hostile, homophobic culture may compromise LGB people’s mental health [4,5].

LGB people can be negatively influenced by sexual stigma not only through the behavioral aspect, such as homophobic harassment and discrimination, but also through the attitudinal aspect. For LGB people, perceived homonegative attitudes in social environments cause minority stress regarding their core identity [6] and endanger their mental health [2,7]. For heterosexual people without a negative attitude toward homosexuality, perceived homonegative attitudes in the general population may disturb them and make them doubt social justice. For heterosexual people with a negative attitude toward homosexuality, perceived homonegative attitudes in the general population may serve as a reason to hold on to their homonegative attitude. Based on these points, examining individuals’ perception of the population’s attitudes toward homosexuality is of great importance to enhancing social acceptance of sexual minorities.

### 1.2. Effects of Demographic Characteristics and Mental Health

A higher level of homosexuality acceptance has been reported in young people than in their older counterparts [8,9,10,11]. Furthermore, homophobic behavior and homonegative attitudes have been more prevalent among heterosexual men than among heterosexual women [12,13]. Research also found attitudes toward same-sex marriage varied across different political and religious grounds [14]. In Taiwan, men were likelier to oppose same-sex marriage than women [15]. Thus, a negative attitude toward LGB rights has been exhibited more by older individuals and men, compared with younger individuals and women [9,10]. However, differences in perception of popular attitudes toward homosexuality across gender and age warrant study.

Over the past two decades, tolerance to homosexuality in Taiwan has outpaced that which is found in China, Japan, and South Korea [8]. However, LGB individuals, who are targets of stigma, may have a lower perception of the general population’s acceptance of homosexuality compared to their heterosexual peers. A positive and accepting attitude toward LGB people was associated with better self-reported health and subjective well-being [16], whereas a low level of support for same-sex marriage was significantly associated with poor mental health [17]. The association between mental health and attitude toward homosexuality was hypothesized to be related to liberal–conservative differences in positive emotion [18], openness to new experiences, uncertainty tolerance, and self-esteem [19] and neurocognitive functioning between conservatives and liberals [20]. Whether mental health is also related to people’s perception of popular attitude toward homosexuality warrants further study.

### 1.3. Role of Social Debates on Legalizing Same-Sex Marriage

A study in Taiwan from 1995 to 2012 claimed that liberal values related to divorce, prostitution, and gender roles acted as mediators for cohort improvement in tolerant attitudes toward homosexuality [9]. However, social debates on LGB human rights may trigger changes in attitudes toward homosexuality in the general population or disclose their attitudes toward homosexuality within a short span. From 2016 to 2018, social debates on legalizing same-sex marriage occurred in Taiwan. In October 2016, a group of legislators introduced a marriage equality bill; however, it failed due to a lack of support from both the ruling and main opposition parties. In May 2017, the Council of Grand Justices ruled that the law barring same-sex marriage was a violation of the human right to equality and was unconstitutional, and it directed that same-sex marriage should be legislated within two years. The opposing group drafted two referenda against the legalization of same-sex marriage, whereas the supporting group drafted a referendum to support the legalization of same-sex marriage. On 24 November 2018, the voting results revealed that 70.12% of voters favored the legalization of same-sex relationship being made outside of changes to the Civil Code; 57.60% favored that “marriage” only applies to couples of different sexes, not to same-sex couples; and only 30.27% supported amending the Civil Code to legalize same-sex marriage.

From 2016 to 2018, the group opposing same-sex marriage placed prominent advertisements on social media and public media to advocate against same-sex marriage and LGB individuals. They claimed that homosexuality is unnatural and abnormal; legalizing same-sex unions would promote homosexuality in society, cause widespread outbreaks of HIV infections, depopulate Taiwan, deteriorate traditional family values; and that children are better off when raised by opposite-sex couples (e.g., https://www.youtube.com/watch?v=MpE87QhI3MQ (accessed on 1 February 2021). Although the supporting group attempted to clarify misleading messages, their claims were rarely heard by the general population due to a lack of funding required to place advertisements on public media.

According to ecological systems theory [21], an individual interacts with environmental systems and is influenced by the values of those same systems. The 2016 to 2018 social debates on legalizing same-sex marriage were the first instances for the people in Taiwan to discuss the issue of homosexuality in public. During these social debates, people might have received and respond to opinions regarding sexuality and sexual minorities from their families, peers, classmates, coworkers, and/or mass media. Thus, social debates on legalizing same-sex marriage could be the factor existing within the microsystem (family, peers, etc.) and the exosystem (mass media, neighbors, industry, etc.) that might influence an individual’s thoughts. Further studies are required to determine whether social debates on legalizing same-sex marriage influence people’s perception of the general population’s attitudes toward homosexuality.

### 1.4. Aims of This Study

The aims of this online survey study were to examine the associations of demographic characteristics (gender, age, and sexual orientation), mental health status, and point in time of the survey (the first month of 2017 vs. the last month of 2018) with people’s perception of the attitudes of the general population in Taiwan toward homosexuality. We conducted the first survey at the beginning of the social debates and the second survey at the end of the social debates. Given that social debates on legalizing same-sex marriage spanned 22 months, we hypothesized that people perceived a lower level of homosexual acceptance from the general population in the second survey compared to the first survey. Furthermore, we hypothesized that gender, age, sexual orientation, and mental health status were related to people’s perceived attitudes toward homosexuality in the general population.

## 2. Methods

### 2.1. Participants

The data were based on the Taiwanese Attitudes toward Homosexuality Survey (TAHS). The first survey (TAHS 2017) was conducted between 1 January and 31 January 2017 (two months after introducing a marriage equality bill by a group of legislators), and the second survey (TAHS 2018) was conducted between 1 December and 31 December 2018 (one week after the same-sex marriage referendum). The method of recruiting participants is described elsewhere [17]. In brief, participants aged at least 20 years were recruited for the online survey through a Facebook advertisement. The Facebook advertisement included a headline, main text, pop-up banner, and weblink to the study questionnaire website. The advertisement appeared in Facebook’s news feed, which is a streaming list of updates from the user’s connections and advertisers. We targeted the advertisement to Facebook users based on location (Taiwan) and language (Chinese). A de-duplication protocol was applied to identify multiple submissions to preserve data integrity, including cross-validation of the eligibility criteria, of key variables and discrepancies in key data, and checking for unusually fast completion time (<10 min) [22]. The study was approved by the Institutional Review Board of Kaohsiung Medical University Hospital. Participants were not given incentives for participation. The study design involved anonymous online responses to the recruitment advertisement and questionnaire, allowing the respondents to freely decide whether to join, and their personal information was secured.

A total of 3437 and 1409 Facebook users completed the online questionnaire in TAHS 2017 and TAHS 2018, respectively. Among them, 137 and 39 were excluded from the analysis because they were underage (<20 years) and provided erroneous values for age (>100 years) in the TAHS 2017 and TAHS 2018, respectively; moreover, 14 were excluded because they responded to the online questionnaire in both the TAHS 2017 and TAHS 2018. Given that the number of transgender respondents was small (47 in TAHS 2017 and 33 in TAHS 2018), the data of transgender respondents were not included for analysis. In total, 3239 and 1323 participants were recruited in the TAHS 2017 and TAHS 2018, respectively.

### 2.2. Measures

#### 2.2.1. Dependent Variable

The dependent variable, namely, perceived attitudes of the general population toward homosexuality, was measured by a single survey item, “To what degree do people in Taiwan accept homosexuality now?” Participants rated the level of perceived homosexuality acceptance on a 5-point Likert scale ranging from 0 (*very low*) to 4 (*very high*).

#### 2.2.2. Independent Variables

The independent variables were gender (female, male, and transgender), age, sexual orientation (heterosexual, homosexual, bisexual, pansexual, asexual, and unsure), mental health status, and point in time of the survey (TAHS 2017 and TAHS 2018). According to sexual orientation, participants were classified into heterosexual and non-heterosexual groups. The non-heterosexual group consisted of homosexuals, bisexuals, and others (pansexual, asexual, and unsure). Respondents’ mental health status was evaluated using the 5-item Brief Symptom Rating Scale (BSRS-5) [21]. The BSRS-5 contains five conditions: (1) feeling tense or keyed up; (2) feeling low in mood; (3) feeling easily annoyed or irritated; (4) feeling inferior to others; and (5) having trouble falling asleep. Respondents were asked to rate the frequency of each condition during the previous week on a 5-point Likert scale from 0 (*not at all*) to 4 (*extremely*). The BSRS-5 has been approved as an efficient tool for the screening of poor mental health prone psychiatric inpatients, general medical patients, and community residents [23,24,25,26,27]. The Cronbach’s alpha was 0.91 in this study. The total score on the BSRS-5 was used as an indicator of mental health. Participants with a total BSRS-5 score of ≥10 were classified as having a poor mental status [23].

### 2.3. Procedure and Statistical Analysis

Gender, age, sexual orientation, mental health status, and perceived attitudes toward homosexuality in the TAHS 2017 and TAHS 2018 were analyzed using descriptive statistics. The association of survey point in time, gender, age, sexual orientation, and mental health with one’s perception of the population’s attitudes toward homosexuality was analyzed using a multiple regression analysis. We examined the assumptions of homoscedasticity and normality of residual using the Breusch–Pagan test and kurtosis and skewness, respectively. If the assumption of homoscedasticity was tenable, the generalized estimating equations with robust estimation were used to address the problem of non-homoscedasticity. For normality of residual, we considered a non-normality if the kurtosis and skewness of the standardized residual were greater than ±2 based on the suggestion of Gravetter and colleagues [28]. A *p* value < 0.05 was considered statistically significant.

## 3. Results

Table 1 presents respondents’ gender, age, sexual orientation, mental health status, and perception of the population’s attitudes toward homosexuality in the TAHS 2017 and TAHS 2018. Respondents who were non-heterosexual (*p* = 0.003) and who had poor mental health (*p* < 0.001) were greater in number in the TAHS 2017 than in the TAHS 2018. Furthermore, respondents in the TAHS 2018 were older and perceived lower homosexuality acceptance than those in the TAHS 2017 (*p* < 0.001).

Table 2 demonstrates the results of multiple regression analysis on the association of survey point in time, gender, age, sexual orientation, and mental health with perceived population attitudes toward homosexuality. Before reporting the results, we examined the homoscedasticity of residual, and based on the Breusch–Pagan test, the assumption of homoscedasticity was not held (*p* < 0.001). Thus, we used the generalized estimating equations with robust estimation to estimate the regression coefficients. For normality of residual, the assumption was held because the skewness (0.06) and kurtosis (0.19) were within ±2. The results indicated that participants’ perception of the population’s acceptance of homosexuality was lower in the TAHS 2018 than in the TAHS 2017. Participants with poor mental health perceived the population’s homosexuality acceptance to be lower than did those with good or fair mental health. Old participants perceived low homosexuality acceptance. No difference was observed between men and women. Non-heterosexual respondents perceived the population’s homosexuality acceptance to be higher than did heterosexual respondents.

We further examined the moderating effects of gender, age, sexual orientation, and mental health on the association between survey point in time and perceived population attitude toward homosexuality (Figure 1). The results indicated that only gender had a moderating effect. Although perceived population homosexuality acceptance among both male and female respondents was lower in the TAHS 2018 than in the TAHS 2017, men exhibited a more significant decline in perceived population homosexuality acceptance than did women.

Table 3 demonstrates the results of multiple regression analysis on the association of gender, age, sexual orientation, and mental health with perceived population’s attitudes toward homosexuality in TAHS 2017 and in TAHS 2018. The results indicated that participants with poor mental health perceived the population’s homosexuality acceptance to be lower than did those with good or fair mental health in TAHS 2017 and TAHS 2018. Non-heterosexual respondents’ perception of the population’s homosexuality acceptance was higher than that of heterosexual respondents in both the TAHS 2017 and TAHS 2018. Male participants perceived the population’s homosexuality acceptance to be lower than females did in the TAHS 2018, whereas no gender difference was found in the TAHS 2017. Older participants perceived the population to have a low homosexuality acceptance in the TAHS 2017, whereas no significant association between age and perceived attitude existed in the TAHS 2018.

## 4. Discussion

### 4.1. Perceived Population Attitudes toward Homosexuality in the TAHS 2017 and TAHS 2018

The World Values Survey, a global research project that explores people’s values and beliefs, how they change over time, and what social and political impact they have, found that in Taiwan, the mean scores on the tolerance measure for whether homosexuality is justified increased by 132% between 1995 and 2012 [8], indicating that attitudes toward homosexuality have become increasingly accepting among people in Taiwan. However, the present study found that peoples’ perception of the population’s attitudes toward homosexuality in Taiwan turned significantly negative from the TAHS 2017 results to those of the TAHS 2018. Given that social debates on legalizing same-sex marriage spanned 22 months between the TAHS 2017 and the TAHS 2018, and that anti-LGB groups had disproportionately stronger financial aid to spread their beliefs compared with LGB-friendly groups, the decline in the perception of the population’s homosexual acceptance may be highly influenced by negative messages spread by the anti-LGB groups during debates. The results supported that as a factor existing in individuals’ microsystem and exosystem, social debates on legalizing same-sex marriage might change people’s perception of the population’s attitudes toward homosexuality.

The government’s role in preventing people from being misled through misinformation against LGB people in social debates is worth noting. In general, the Taiwanese government kept silent during the debates. According to the Referendum Act, the Taiwanese government should present official opinions regarding the referenda in nationally televised debates held by the Central Election Commission; however, the Taiwanese government was absent in all nationally televised debates and did not attempt to clarify misinformation spread by anti-LGB groups. Moreover, LGB-supporting groups reported anti-LGB advertisements to the National Communications Commission (NCC), claiming they were spreading misinformation regarding homosexuality, and although the NCC agreed that the advertisements contained misinformation, it did not provide clear information in a timely manner for the public to take into consideration [29].

The results indicated that determining the civil rights of sexual minority individuals through referenda may harm sexual minorities. Ways to prevent damage to the image of minority groups resulting from misinformation in social debates on minority group rights must be studied in depth. Although the Council of Grand Justices ruled that same-sex marriage should be legislated within two years, the Taiwanese government announced the Act for Implementation of Judicial Yuan Interpretation No. 748 outside the Civil Code in May 2019 based on the results of the two referenda against same-sex marriage. This law guaranteed most, but not all, of the same rights entailed in a heterosexual marriage for same-sex couples and deprived the possibility of marriage for same-sex couples according to the Civil Code. Whether the legalization of same-sex relationships, but not same-sex marriage, can change people’s attitudes toward homosexuality warrants inspection.

### 4.2. Demographic Characteristics and Mental Health

The present study demonstrated that older participants perceived lower homosexuality acceptance in the TAHS 2017. Older people have shown a lower level of homosexuality acceptance than their younger counterparts [8,9,10,11]. Older people may perceive the population’s homonegative attitudes from interaction with people of the same generation and simultaneously reinforce their own negative attitude toward homosexuality. The present study did not find gender difference in the perceived population’s attitude toward homosexuality in the TAHS 2017, whereas men perceived the population’s homosexuality acceptance to be lower than women did in the TAHS 2018; moreover, men had a significant decline in perceived population homosexuality acceptance compared with women between the TAHS 2017 and the TAHS 2018. This result indicated that social debates on legalizing same-sex marriage between the TAHS 2017 and TAHS 2018 influenced men more than women. The number of transgender participants in the present study was small, and the perceived population’s attitudes toward homosexuality warrants further study utilizing a larger sample of transgender participants.

Contrary to our hypothesis, the present study found that non-heterosexual respondents perceived a higher level of homosexuality acceptance in the population than did heterosexual respondents. One of the possible etiologies accounting for the result is the social interaction spectrum. Due to the social stigma regarding homosexuality, many LGB people do not reveal their sexual orientation to others. Most heterosexual people in Taiwan have little chance to interact with non-heterosexual ones, and thus, traditional homonegative attitudes may be persistently maintained by many heterosexual people. To increase the possibility of heterosexual people interacting with LGB individuals, and introducing unbiased viewpoints of homosexuality to them, sustained efforts are required.

Additionally, the present study indicated that participants with poor mental health perceived lower homosexuality acceptance in the general population than did those who had good or fair mental health. The cross-sectional study design limited the possibility of determining the causal relationship between perceived homonegative attitude and poor mental health in the general population. Perception of the population’s homonegative attitude may negatively affect the mental health of people who have a friendly attitude toward LGB people. On the other hand, poor mental health may make people focus on the population’s negative aspects of attitudes toward homosexuality. It is also possible that people with poor mental health might look forward to social resources to improve their lives; however, the debate on same-sex marriage had drained the resources and attention of the society. As a result, people with poor mental health might hold a more negative view toward what was debated in the society.

Given that the BSRS-5 contains an item assessing “feeling inferior to others”, we further examined the correlation between the level of feeling inferior to others and the perceived population’s attitude toward homosexuality among heterosexual and non-heterosexual respondents. The results demonstrated that a higher level of feeling inferior to others significantly correlated with a perceived higher homonegative attitude among non-heterosexual respondents in both the TAHS 2017 (Pearson’s correlation *r* = −0.098, *p* < 0.001) and the TAHS 2018 (*r* = −0.102, *p* = 0.003), whereas no significant correlation was found in heterosexual individuals in the TAHS 2017 (*r* = −0.039, *p* = 0.135) or TAHS 2018 (*r* = 0.004, *p* = 0.935). It is possible that non-heterosexual individuals may integrate the thought of the population’s negative attitude toward homosexuality into their opinion of themselves.

### 4.3. Limitations

This study had several limitations. First, we recruited participants through Facebook advertisements; therefore, our sample was not nationally representative. Second, we enrolled two different samples in the TAHS 2017 and TAHS 2018 because anyone who saw the Facebook advertisement could join our study. We only assessed the changes in perceived population attitude toward homosexuality between the TAHS 2017 and TAHS 2018 at the population level, not at the personal level. This limited the possibility of determining the causal effects of debates on legalizing same-sex marriage on the population’s attitude toward homosexuality. Third, further studies are needed to examine the associations of education level, experience of interacting with LGB people, and political and religious grounds with perceived population attitude toward homosexuality. Fourth, we evaluated the respondents’ mental health status using the BSRS-5. Although the BSRS-5 has been approved as an efficient tool for the screening of poor mental health prone psychiatric inpatients, general medical patients, and community residents [23,24,25,26,27], it did not comprehensively evaluate participants’ symptoms of psychiatric illnesses. Therefore, the present study could not determine which psychiatric symptoms are significantly associated with perceived population attitudes toward homosexuality. Fifth, although using a single item to measure perceived attitude is a practical and feasible way in collecting data online, the single item measure had lower validity than a scale-like measure.

## 5. Conclusions

The perceived population attitude toward homosexuality varied among groups of people with different genders, ages, sexual orientations, and mental health statuses. Perceived population attitude toward homosexuality in the TAHS 2018 was significantly lower than that in the TAHS 2017, indicating that negative messages spread by the anti-LGB groups during debates on legalizing same-sex marriage over the period of 22 months between the TAHS 2017 and TAHS 2018 might have negatively influenced perceived homosexual acceptance in the general population. Appropriate inspections must be conducted to prevent misleading the population regarding homosexuality in social debates on LGB people’s human rights.

## Figures and Tables

**Figure 1 ijerph-18-02618-f001:**
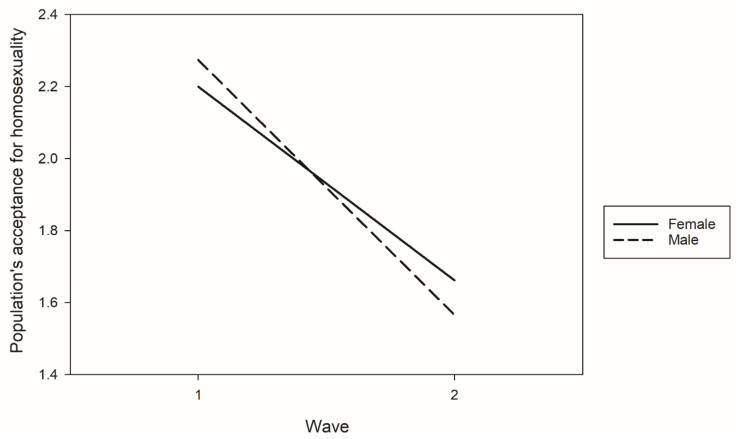
Gender differences in perceived homosexuality acceptance in the general population in the two surveys.

**Table 1 ijerph-18-02618-t001:** Gender, sexual orientation, mental health status, age, and perceived attitudes toward homosexuality in TAHS (Taiwanese Attitudes toward Homosexuality Survey) 2017 and TAHS 2018.

Variable	TAHS 2017 (*n* = 3239)	TAHS 2018 (*n* = 1323)
	*n* (%)	*n* (%)
**Gender**		
Female	2049 (63.26)	819 (61.90)
Male	1190 (36.74)	504 (38.10)
**Sexual orientation**		
Heterosexuality	1443 (44.55)	535 (40.44)
Nonheterosexuality	1796 (55.45)	788 (59.56)
**Mental health**		
Good or fair	2603 (80.36)	868 (65.61)
Poor	636 (19.64)	455 (34.39)
	Mean (SD)	Mean (SD)
**Age** (years)	30.59 (8.06)	32.20 (9.17)
Acceptance for homosexuality	2.23 (0.86)	1.63 (0.85)

**Table 2 ijerph-18-02618-t002:** Association of survey year, gender, sexual orientation, mental health, and age with perceived attitudes toward homosexuality: multiple regression analysis.

Variables	Adjusted Regression Coefficient	*p*-Value
TAHS 2018 (reference = TAHS 2017)	−0.571	<0.001
Gender: Male (reference = Female)	−0.010	0.723
Nonheterosexuality (reference = heterosexuality)	0.147	<0.001
Poor mental health (reference = good or fair mental health)	−0.166	<0.0001
Age	−0.008	<0.0001

**Table 3 ijerph-18-02618-t003:** Association of gender, sexual orientation, mental health, and age with perceived attitudes toward homosexuality in TAHS 2017 and TAHS 2018: multiple regression analysis.

Variables	TAHS 2017		TAHS 2018	
	Adjusted Regression Coefficient	*p*-Value	Adjusted Regression Coefficient	*p*-Value
Gender: Male (reference = Female)	−0.0211	0.190	−0.241	0.004
Nonheterosexuality (reference = heterosexuality)	0.0644	<0.001	0.098	<0.001
Poor mental health (reference = good or fair mental health)	−0.2529	<0.001	−0.241	0.007
Age	−0.0137	<0.001	−0.008	0.402

## Data Availability

Restrictions apply to the availability of these data. Only researchers of this study can access the data.

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
