# Peer review of "Perception of Attitudes of the General Population toward Homosexuality in Taiwan: Roles of Demographic Factors, Mental Health, and Social Debates on Legalizing Same-Sex Marriage"

_ijerph, 2021, doi:10.3390/ijerph18052618_

Round 1
Reviewer 1 Report
One assumption in the paper is that people discriminate against LGBT persons because of their sexual orientation. In part perhaps, but I suspect if it were studied, one would find that much of the discrimination is on account of gender unusual behavior, hair coloring, gait, and other things associated with some LGBT persons but not sexual orientation per se (as in why should I care about what someone does in their own bedroom, but I might care if you color your hair green when the local team colors are red).
Sexual minority theory is ok as far as it goes but I think a deeper look might find more. When it comes to social justice, an equity perspective might lead one to be against same-sex sexual activity in many of its forms (not all). For example, suppose one assumes that many people assume some degree of risk and cost when they marry (no matter the gender) by giving up sex with others. Well, if someone comes along and says that they want to be married but keep having sex with others (regardless of their gender or sexual orientation), those who are married without having sex with others might feel it was unjust for both groups to get equal legal and social benefits. One way society may reestablish equity is to allow for the underbenefitted to criticize the overbenefitted in this case, using social exchange theory.
I think it was Schumm's paper on forty years of controversial research (comprehensive psychology, 2015?) where he found that what predicted support for same-sex marriage and parenting was less being a Democrat, highly educated, female, or liberal but believing in casual sex as morally neutral or good. As discussed in paragraph 1.3.
In the last paragraph in section 1.3 on page 3, I'd suggest being a bit less dogmatic. In Schumm's 2018 book (same-sex parenting research) he did find that when U.S. states approved of same-sex marriage earlier, they did have later ages at marriage and lower fertility rates; not quite depopulating the USA but the trend was in that direction. As far as children being better/worse off, there is increasing evidence that children of same-sex couples are more likely to grow up to be LGBT (see Gartrell et al., 2019, 70% rate for daughters), some evidence they are more likely to use illegal drugs (Gartrell et al., 2015), and that their parental relationships are less stable (Allen and Price, 2018, Marriage and Family Review). Gates (2015) also noted the same instability as leading to problems in the lives of their children. It might not be lack of money but reluctance to engage in a debate that might bring some of the scientific data to the fore that could have explained why the debate in Taiwan seemed one sided.
The most serious problem with this paper is that non-representative sample; not just that it was done on facebook. Over half of the surveys at both times were filled out by nonheterosexuals. That might be great for catching the views of nonheterosexuals but I wonder about how representative were the views of heterosexuals. Furthermore, the data almost begs for a prediction of mental health status at each time from gender, transgender, LGB, and age, and perceived discrimination (if that is available).
Did the authors look at the two-way interaction effect of mental health and sexual orientation on the dependent variable? One might predict that LGB/T persons with poorer mental health might see the situation differently than LGB/T persons with better mental health.
I am not familiar with the negative messages spread by anti-LGB groups in Taiwan. However, the use of such language seems less scientific to me since it seems to assume as a matter of fact that such groups and their messages were factually incorrect or biased by negative emotion. Perhaps they were, but I'd prefer to see more evidence.
I always get suspicious when authors refuse to release their data for others to independently check. It's risky - sure Regnerus got some severe criticism when others took his free data and reanalyzed it, but he did the honest thing even if it cost him.
Author Response
Comment 1 One assumption in the paper is that people discriminate against LGBT persons because of their sexual orientation. In part perhaps, but I suspect if it were studied, one would find that much of the discrimination is on account of gender unusual behavior, hair coloring, gait, and other things associated with some LGBT persons but not sexual orientation per se (as in why should I care about what someone does in their own bedroom, but I might care if you color your hair green when the local team colors are red). Response Thank you for your comment. Given that there have been many studies demonstrating that discrimination against LGBT persons is prevalent around the world, we took the perspective of previous studies. Comment 2 Sexual minority theory is ok as far as it goes but I think a deeper look might find more. When it comes to social justice, an equity perspective might lead one to be against same-sex sexual activity in many of its forms (not all). For example, suppose one assumes that many people assume some degree of risk and cost when they marry (no matter the gender) by giving up sex with others. Well, if someone comes along and says that they want to be married but keep having sex with others (regardless of their gender or sexual orientation), those who are married without having sex with others might feel it was unjust for both groups to get equal legal and social benefits. One way society may reestablish equity is to allow for the underbenefitted to criticize the overbenefitted in this case, using social exchange theory. Response Thank you for your suggestion. The present study did not examine the value of same-sex marriage. We will take your suggestion into consideration if we examine the social value of same-sex marriage in future. Comment 3 I think it was Schumm's paper on forty years of controversial research (comprehensive psychology, 2015?) where he found that what predicted support for same-sex marriage and parenting was less being a Democrat, highly educated, female, or liberal but believing in casual sex as morally neutral or good. As discussed in paragraph 1.3. Response Although we could not find the paper you mentioned, we found similar studies that examined the relationships between political and religious grounds and attitudes toward same-sex marriage as below. We also added political and religious grounds as the topics that need further study on their association with perceived population’s attitude toward homosexuality. In Introduction section “Research also found attitudes toward same-sex marriage varied across different political and religious grounds [14].” In Discussion section “Third, further studies are needed to examine the associations of education level, experience of interacting with LGB people, political and religious grounds with perceived population’s attitude toward homosexuality.” Comment 4 In the last paragraph in section 1.3 on page 3, I'd suggest being a bit less dogmatic. In Schumm's 2018 book (same-sex parenting research) he did find that when U.S. states approved of same-sex marriage earlier, they did have later ages at marriage and lower fertility rates; not quite depopulating the USA but the trend was in that direction. As far as children being better/worse off, there is increasing evidence that children of same-sex couples are more likely to grow up to be LGBT (see Gartrell et al., 2019, 70% rate for daughters), some evidence they are more likely to use illegal drugs (Gartrell et al., 2015), and that their parental relationships are less stable (Allen and Price, 2018, Marriage and Family Review). Gates (2015) also noted the same instability as leading to problems in the lives of their children. It might not be lack of money but reluctance to engage in a debate that might bring some of the scientific data to the fore that could have explained why the debate in Taiwan seemed one sided. Response Thank you for your comment. Given that the law did not allow the same-sex couples to rear children in Taiwan before 2019, there are very few children of same-sex couples during the period of this study because same-sex marriage. Therefore, the children of same-sex couples did not have role in this debate in Taiwan. However, it is really important to do the research on the development of children reared by same-sex couples in Taiwan. Comment 5 The most serious problem with this paper is that non-representative sample; not just that it was done on facebook. Over half of the surveys at both times were filled out by nonheterosexuals. That might be great for catching the views of nonheterosexuals but I wonder about how representative were the views of heterosexuals. Furthermore, the data almost begs for a prediction of mental health status at each time from gender, transgender, LGB, and age, and perceived discrimination (if that is available). Response We agreed that the participants of this study were not a nationally representative sample. We have listed it as the first limitations of this study. Please refer to 4.3. Limitations section. Comment 6 Did the authors look at the two-way interaction effect of mental health and sexual orientation on the dependent variable? One might predict that LGB/T persons with poorer mental health might see the situation differently than LGB/T persons with better mental health. Response Thank you for your suggestion. We found that the interaction between sexual orientation and mental health was not significantly associated with perceived population’s attitude toward homosexuality (p = 0.950), indicating that mental health did not moderate the association between sexual orientation and perceived population’s attitude toward hososexuality. Comment 7 I am not familiar with the negative messages spread by anti-LGB groups in Taiwan. However, the use of such language seems less scientific to me since it seems to assume as a matter of fact that such groups and their messages were factually incorrect or biased by negative emotion. Perhaps they were, but I'd prefer to see more evidence. Response We added the link to one of the television advertisement made by anti-LGB groups into the revised manuscript for your reference. The National Communications Commission has judged that this advertisement spread misinformation regarding homosexuality [as reference 22]. Comment 8 I always get suspicious when authors refuse to release their data for others to independently check. It's risky - sure Regnerus got some severe criticism when others took his free data and reanalyzed it, but he did the honest thing even if it cost him. Response We are glad to provide the data for analysis if International Journal of Environmental Research and Public Health requests, just like what the journal PLOS One requests.
Reviewer 2 Report
I would like to congratulate the authors for this work. I think it is an interesting study with important findings, although I think that it can be improved theoretically and methodologically. These are my suggestions:
- The authors use the Bronfenbrenner model at the beginning of 1.2 and 1.3 sections. However, they do not clearly explain the model and where the predictors are located within this approach. In other words, are the predictors part of the microsystem, mesosystem, exosystem, or macrosystem? How are these systems defined? Additionally, there are no references to this model in the discussion, so the readers cannot know how these findings advance theoretical knowledge on this model.
- PARTICIPANTS: How did the authors control that the sincerity in the responses (apart from what they wrote in the second paragraph of the participants' section? Did they control the time of responding the whole questionnaire?
- MEASURES AND TITLE: The dependent variable is measured with only an item, therefore the reliability of this measure is compromised. Why did not the authors use a scale-like measure? More importantly, the item used is measuring the people's perception of their fellow citizens (To what degree do people in Taiwan accept homosexuality now?), but not their own attitudes. Why did not the authors asked about their own people's attitudes instead of their perception of others? I think this would have made more precise the effect of the social debates.
- RELIABILITY: Only one item to measure the dependent variable and no report of reliability for the BSRS-5. In this last case, Cronbach's alpha should be reported.
- STATISTICAL ANALYSES: Did the authors think to use a multilevel approach with such a big sample? Why did they choose this approach instead?
- RESULTS: My biggest concern is that the main results are using a very short sample for transgender (47 in 2017 and 33 in 2018) and a huge sample for male and female (bigger than a thousand). The authors should revise the assumptions to run multiple regression analyses and report them (multivariate normality, heteroscedasticity, etc.).
Again, congratulations on this work. I mainly make some questions to the authors to think in different ways to improve the quality of the final manuscript, although I recognize important merits in their work.
Author Response
Comment 1
The authors use the Bronfenbrenner model at the beginning of 1.2 and 1.3 sections. However, they do not clearly explain the model and where the predictors are located within this approach. In other words, are the predictors part of the microsystem, mesosystem, exosystem, or macrosystem? How are these systems defined? Additionally, there are no references to this model in the discussion, so the readers cannot know how these findings advance theoretical knowledge on this model.
Response
Thank you for your reminding. In the revised manuscript we added a paragraph regarding the role of social debates on legalizing same-sex marriage in Bronfenbrenner ecological systems model as below.
In Introduction section:
“According to ecological systems theory [21], an individual interacts with environmental systems and is influenced by the values of the environmental systems. The social debates on legalizing same-sex marriage is the first time for the people in Taiwan to discuss the issue of homosexuality in public. People might receive and respond to the opinions regarding sexuality minority from their families, peers, classmates, coworkers, and mass media during the period of social debates. Thus, social debates on legalizing same-sex marriage could be the factor existing in microsystem and exosystem that might influence the individuals’ thought within and outer lives. Further studies are required to determine whether social debates on legalizing same-sex marriage influence people’s perception of attitudes toward homosexuality in the general population.”
In Discussion section:
“The result supported that as a factor existing in individuals’ microsystem and exosystem, social debates on legalizing same-sex marriage might change people’s perception of population’s attitudes toward homosexuality.”
Comment 2
PARTICIPANTS: How did the authors control that the sincerity in the responses (apart from what they wrote in the second paragraph of the participants' section? Did they control the time of responding the whole questionnaire?
Response
Yes, we applied the protocol to preserve data integrity. We added the protocol as below into the revised manuscript.
“A de-duplication protocol was applied to identify multiple submissions to preserve data integrity, including cross-validation of the eligibility criteria of key variables and discrepancies in key data and checking for unusually fast completion time (< 10 minutes) [22].”
Comment 3
MEASURES AND TITLE:
- The dependent variable is measured with only an item, therefore the reliability of this measure is compromised. Why did not the authors use a scale-like measure?
- More importantly, the item used is measuring the people's perception of their fellow citizens (To what degree do people in Taiwan accept homosexuality now?), but not their own attitudes. Why did not the authors asked about their own people's attitudes instead of their perception of others? I think this would have made more precise the effect of the social debates.
Response
- We agreed the reviewer’s comment. Although using single item is a practical and feasible way collect data online, only an item had lower validity than a scale-like measure. We listed it as one of the limitations in this study as below.
“Fifth, although using single item to measure perceived attitude is a practical and feasible way in collecting data online, only an item had lower validity than a scale-like measure.”
- Thank you for your comment. We described the importance of examining individuals’ perception of population’s attitude toward homosexuality as below in Introduction section.
“For LGB people, perceived homonegative attitudes in social environments cause minority stress regarding their core identity [6] and endanger their mental health [2,7]. For heterosexual people without a negative attitude toward homogeneity, perceived homonegative attitudes in the general population may disturb them and make them doubt social justice. For heterosexual people with a negative attitude toward homosexuality, perceived homonegative attitudes in the general population may serve as a reason to hold on to their homonegative attitude. Based on these points, examining individuals’ perception of population’s attitudes toward homosexuality is of great importance to enhancing social acceptance of sexual minority.”
Comment 4
RELIABILITY: No report of reliability for the BSRS-5. In this last case, Cronbach's alpha should be reported.
Response
Thank you for your reminding. We added more introduction of the BSRS-5 and its Cronbach's alpha as below into the revised manuscript.
“The BSRS-5 has been approved to be an efficient tool for the screening of poor mental health-prone psychiatric inpatients, general medical patients, and community residents [23-27]. The Cronbach’s alpha was 0.91 in this study.”
Comment 5
STATISTICAL ANALYSES: Did the authors think to use a multilevel approach with such a big sample? Why did they choose this approach instead?
Response
Thank you for your suggestion. In addition to the original analysis, we divided the participants into those in TAHS 2017 and in TAHS 2018 and examined the association of gender, age, sexual orientation, and mental health with perceived population’s attitudes toward homosexuality in different timepoints. We described the results in Table 3 and the last paragraph of the Results section (as below). We also revised the contents of Discussion section based on the new analysis results.
In Abstract section
“The results also indicated that poor mental health and heterosexual orientation were significantly associated with low perceived population’s homosexuality acceptance in both two waves of survey.”
In Results section
“Table 3 demonstrates the results of multiple regression analysis on the association of gender, age, sexual orientation, and mental health with perceived population’s attitudes toward homosexuality in TAHS 2017 and in TAHS 2018. The results indicated that participants with poor mental health perceived the population’s homosexuality acceptance to be lower than did those with good or fair mental health in TAHS 2017 and TAHS 2018. Nonheterosexual respondents perceived population’s homosexuality acceptance to be higher than did heterosexual respondents in TAHS 2017 and TAHS 2018. Male participants perceived population’s homosexuality acceptance to be lower than did females in TAHS 2018, whereas no difference was found between male and female participants in TAHS 2017. Old participants perceived low homosexuality acceptance in TAHS 2017, whereas no significant association between age and perceived attitude existed in TAHS 2018.”
In Discussion section
“The present study demonstrated that older participants perceived lower homosexuality acceptance in TAHS 2017.”
“The present study did not found gender difference in perceived population’s attitude toward homosexuality in TAHS 2017, whereas men perceived population’s homosexuality acceptance to be lower than did women in TAHS 2018.”
Comment 6
RESULTS:
- My biggest concern is that the main results are using a very short sample for transgender (47 in 2017 and 33 in 2018) and a huge sample for male and female (bigger than a thousand).
- The authors should revise the assumptions to run multiple regression analyses and report them (multivariate normality, heteroscedasticity, etc.).
Response
- Thank you for your comment. The number of transgender respondents was really small. In the revised manuscript we only included the data of female and male respondents for analysis. The significant predictors found in the re-analysis were the same as the original analysis.
“Given that the number of transgender respondents was small (47 in TAHS 2017 and 33 in TAHS 2018), the data of transgender respondents were not included for analysis. In total, 3,239 and 1,323 participants were recruited in the TAHS 2017 and TAHS 2018, respectively.”
- We added the report for multivariate normality and heteroscedasticity as below in the revised manuscript.
In Methods section
“We examined the assumptions of homoscedasticity and normality of residual by using the Breusch–Pagan test and kurtosis and skewness, respectively. If the assumption of homoscedasticity was tenable, the generalized estimating equations with robust estimation was used to address the problem of non-homoscedasticity. For normality of residual, we considered a non-normality if the kurtosis and skewness of the standardized residual were greater than ± 2 based on the suggestion of Gravetter and colleagues [28].”
In Results section
“Before reporting the results, we examined the homoscedasticity of residual, and based on the Breusch–Pagan test, the assumption of homoscedasticity was not held (P < 0.001). Thus, we used the generalized estimating equations with robust estimation to estimate the regression coefficients. For normality of residual, the assumption was held because the skewness (0.06) and kurtosis (0.19) were within ± 2.”
Comment 7
Again, congratulations on this work. I mainly make some questions to the authors to think in different ways to improve the quality of the final manuscript, although I recognize important merits in their work.
Response
Thank you for your comment.
Reviewer 3 Report
I have reviewed this paper and have the following issues for the authors to address.
1) The study surveyed general population using a brief scale, BSRS-5. This scale is very brief and the authors should state this as a limitation as longer questionnaires seem to be more valid to classify people into good and poor mental health.
2) The authors should discuss why poor mental health people had a more negative attitude towards homosexuality. It could be that they faced other social problems in life and the debate on same-sex marriage had drained the resources of the society. As a result, they held a more negative view as compared to one year ago. Please include this point in the discussion.
3) The authors stated "LGB-supporting groups reported against the advertisements spreading misinformation regarding homosexuality placed by anti-LGB groups to the National Communications Commission (NCC); however, the NCC did not punish the groups that paid to advertise based on the reason that “the Taiwanese society needs reconciliation and inclusion". Besides, the government, the authors of a paper need to neutral on this topic. In a debate, both sides would not agree on the views on the opposite sides. The authors claim the side against homosexuality introduced false information to society. Similarly, the opposite camp would say the same thing. Other countries like Australia went through such a debate and the content of advertisement is to devalidate the arguments of other sides. I recommend the authors to remove the above statement and be neutral throughout the paper.
Author Response
Comment 1
The study surveyed general population using a brief scale, BSRS-5. This scale is very brief and the authors should state this as a limitation as longer questionnaires seem to be more valid to classify people into good and poor mental health.
Response
Thank you for your comment. We added it as one of the limitations in this study as below.
“Fourth, we evaluated the respondents’ mental health status using the BSRS-5. Although the BSRS-5 has been approved to be an efficient tool for the screening of poor mental health-prone psychiatric inpatients, general medical patients, and community residents [23-27], it did not comprehensively evaluate participants’ symptoms of psychiatric illnesses. Therefore, the present study could not determine which psychiatric symptoms significantly associate with perceived population’s attitudes toward homosexuality.”
Comment 2
The authors should discuss why poor mental health people had a more negative attitude towards homosexuality. It could be that they faced other social problems in life and the debate on same-sex marriage had drained the resources of the society. As a result, they held a more negative view as compared to one year ago. Please include this point in the discussion.
Response
Thank you for your suggestion. We include this point in the discussion as below.
“It is also possible that people with poor mental health might look forward to social resources to improve their live; however, the debate on same-sex marriage had drained the resources and attention of the society. As a result, people with poor mental health might hold a more negative view toward what were debated in the society.”
Comment 3
The authors stated "LGB-supporting groups reported against the advertisements spreading misinformation regarding homosexuality placed by anti-LGB groups to the National Communications Commission (NCC); however, the NCC did not punish the groups that paid to advertise based on the reason that “the Taiwanese society needs reconciliation and inclusion". Besides, the government, the authors of a paper need to neutral on this topic. In a debate, both sides would not agree on the views on the opposite sides. The authors claim the side against homosexuality introduced false information to society. Similarly, the opposite camp would say the same thing. Other countries like Australia went through such a debate and the content of advertisement is to devalidate the arguments of other sides. I recommend the authors to remove the above statement and be neutral throughout the paper.
Response
Thank you for your suggestion. We deleted this sentence and revised the paragraph as below.
“Although the NCC agreed that the advertisements had misinformation, the NCC did not clear up the misinformation in time for the public [22].”
Round 2
Reviewer 1 Report
The English needs improvement in several places.
On page 8, the finding contrary to their expectations could use further analysis. The scale used to assess mental health contained an item "feeling inferior to others" which could be a marker for internalized homophobia among LGB persons. It would be of interest to see if the relationship between LGB status and the dependent variable differed as a function of the responses to that item. In other words if you're LGB and you feel inferior to others would that lead to a different model than if you're LGB and you don't feel inferior to others? It's about as close as you could get to exploring the role of internalized homophobia without actually having measured it directly.
Author Response
Comment
The English needs improvement in several places.
Response
We invited another English native editor to edit the revised manuscript. We also attached the certificate of edition.
Comment
On page 8, the finding contrary to their expectations could use further analysis. The scale used to assess mental health contained an item "feeling inferior to others" which could be a marker for internalized homophobia among LGB persons. It would be of interest to see if the relationship between LGB status and the dependent variable differed as a function of the responses to that item. In other words if you're LGB and you feel inferior to others would that lead to a different model than if you're LGB and you don't feel inferior to others? It's about as close as you could get to exploring the role of internalized homophobia without actually having measured it directly.
Response
Thank you for your suggestion. We analyzed the data and added the results into the revised manuscript as below.
“Given that the BSRS-5 contains an item assessing “feeling inferior to others,” we further examined the correlation between the level of feeling inferior to others and the perceived population’s attitude toward homosexuality among heterosexual and non-heterosexual respondents. The results demonstrated that a higher level of feeling inferior to others significantly correlated with a perceived higher homonegative attitude among non-heterosexual respondents in both the TAHS 2017 (Pearson’s correlation r = -0.098, p < 0.001) and the TAHS 2018 (r = -0.102, p = 0.003), whereas no significant correlation was found in heterosexual individuals in the TAHS 2017 (r = -0.039, p = 0.135) or TAHS 2018 (r = 0.004, p = 0.935). It is possible that non-heterosexual individuals may integrate the thought of the population’s negative attitude toward homosexuality into the opinion of themselves.”